# The Impact of Postoperative Radiotherapy on Dietary Function of Head and Neck Cancer Patients after Pharyngoesophageal Reconstruction with Free Jejunal Flap

**DOI:** 10.3390/jcm11102860

**Published:** 2022-05-19

**Authors:** Lan Sook Chang, Hyun Wang, Hee Chang Ahn, Tae Hyeon Lee, Kyung Tae, Seong Oh Park

**Affiliations:** 1Department of Plastic and Reconstructive Surgery, Hanyang University College of Medicine, Seoul 04763, Korea; changls02@gmail.com (L.S.C.); wanghyunie@naver.com (H.W.); lthh1@naver.com (T.H.L.); 2Department of Plastic and Reconstructive Surgery, CHA University Bundang Medical Center, Seongnam-si 13496, Korea; ahnhc12@gmail.com; 3Department of Otolaryngology-Head and Neck Surgery, Hanyang University College of Medicine, Seoul 04763, Korea; kytae@hanyang.ac.kr

**Keywords:** dietary function, PORT, jejunal flap, dysphagia, pharyngoesophageal reconstruction

## Abstract

This study investigated the impact of postoperative radiotherapy (PORT) on dietary function in patients who underwent pharyngoesophageal defect reconstruction using a free jejunal flap. A retrospective chart review of 36 patients who underwent circumferential pharyngoesophageal defect reconstruction using a free jejunal flap was performed. The European Organization for the Research and Treatment of Cancer head and neck cancer module questionnaire was used. Five items related to dietary function were selected and analyzed for changes in scores before and after PORT. Both the PORT and non-PORT groups showed improved dietary function before surgery, and no significant changes were noted at 3 and 12 months postoperatively. Repeated measures ANOVA showed that PORT had no significant impact on dietary function. In univariate analysis, no variable was a significant predictor of the score at 12 months, postoperatively. Previous radiotherapy and neck dissection had a close statistically significant relation. The multivariate analysis showed that neoadjuvant chemotherapy, previous radiotherapy, and neck dissection were significant predictors of the score at 12 months, postoperatively. PORT did not show a significant effect on the 12 months postoperative score. Free jejunal flap is an effective pharyngoesophageal defect reconstruction method that does not cause any dietary function disruption after PORT.

## 1. Introduction

One of the most commonly used flaps for circumferential pharyngoesophageal reconstruction is the free jejunal flap. The jejunal flap is generally regarded to have many advantages, such as a low fistula rate, fast primary healing, and superior swallowing function [1]. However, it also has some disadvantages, such as high donor site morbidity and high stricture rate [2]. The evaluation method differs between various studies, and many of them showed improved results with the accumulation of experience. Therefore, it is difficult to definitively conclude what the benefits and pitfalls of the jejunal flap transfer technique are [1,3].

Most patients with head and neck cancer require radiation therapy after tumor resection, which can affect the transferred flap. Therefore, the effect of radiation therapy on the flap can be an important factor influencing flap choice. The jejunum is known to be radiosensitive, and radiation injury of the gastrointestinal tract is well-known in patients with pelvic or colorectal malignancies who undergo external beam irradiation. In previous studies, Peter et al. revealed that radiation doses >45 Gy are toxic to the small bowel [4], and Baglan et al. found that there is a strong dose–volume relationship between radiation doses and small bowel toxicity [5]. Usually, patients undergoing surgery for squamous cell carcinoma of the head and neck regions are exposed to postoperative radiotherapy (PORT) at doses from 57 to 63 Gy. If a transferred jejunal flap is injured due to radiation therapy, stricture and dysphagia may occur. This, in turn, may have a detrimental effect on the patient’s dietary function, which is one of the most important functional outcomes of pharyngoesophageal reconstruction.

There are many studies on the differences between, and the benefits, pitfalls, and functional outcomes of the jejunal and fasciocutaneous flaps [3]. However, to our knowledge, there is a paucity of studies focusing on the radiation effect. Therefore, the aim of this study was to evaluate the effect of PORT on dietary function in patients undergoing circumferential pharyngoesophageal reconstruction using a free jejunal flap. We hypothesized that there is no difference in dietary function between the patients who underwent PORT and those who did not.

## 2. Materials and Methods

A retrospective chart review was performed for all patients who underwent circumferential pharyngoesophageal defect reconstruction using a free jejunal flap from 2002 to 2016. All reconstruction procedures were performed by a single experienced surgeon. Patients with insufficient medical data or a follow-up duration of <2 years were excluded. Patients with tumor invasion to the tongue or mouth floor, which can have a detrimental effect on swallowing function and thereby dietary function, were excluded.

The baseline characteristics of the patients and information regarding cancer, surgery, previous radiation therapy, neoadjuvant chemotherapy, postoperative chemotherapy, and PORT were reviewed. Moreover, complications, esophagography, time period from surgery to oral intake, and whether the patient underwent endoscopic balloon dilatation were also reviewed. Dietary function was evaluated using the European Organization for the Research and Treatment of Cancer (EORTC) head and neck cancer module (QLQ-H&N35) [6]. It comprises a total of 35 items, and patients can choose between 1 and 4 points (not at all–very much) for every question. Five items related to dietary function were selected and analyzed for changes in scores before and after PORT. The patients were asked to complete the questionnaire after 3 months (1 month post-PORT) and 12 months postoperatively (Table 1).

For radiation effect evaluation, patients were divided into two groups: those who underwent PORT (PORT+) and those who did not (PORT−). The Mann–Whitney U test was performed to analyze continuous variables, while the Fisher’s exact and chi-squared tests were used to analyze categorical variables. A repeated measures ANOVA test was performed to compare dietary function between the two groups. To evaluate the internal consistency of the questionnaire, Cronbach’s alpha coefficient was determined. To evaluate factors influencing dietary function, univariate and multivariate regression analyses were performed. The target value was set as the 12-month postoperative questionnaire score. A *p* value of <0.05 was considered statistically significant. Statistical analysis was performed using SAS version 9.4 (SAS Institute Inc., Cary, NC, USA).

## 3. Results

### 3.1. General Characterictics

A total of 36 patients were included in the study. The mean age at operation was 55.22 years (range, 35–82 years). Thirty-four patients were male, and two patients were female. Among the 36 patients, 28 (77.8%) had hypopharyngeal cancer, 4 (11.1%) had upper esophageal cancer, and 4 (11.1%) had laryngeal cancer. Furthermore, 25 (69.4%) patients had primary cancer and 11 (30.6%) had recurrence. All patients underwent total pharyngolaryngectomy with tracheostomy. There were 11 (30.6%) patients who had undergone previous radiation therapy, while 25 (69.4%) had undergone neoadjuvant chemotherapy. Adjuvant chemotherapy was performed in 18 patients (50.0%), and PORT was performed in 22 patients (61.1%). The mean follow-up duration was 49.47 months (range, 28–155 months). Only adjuvant chemotherapy showed statistically significant difference between the two groups (Table 2).

Concomitant neck dissection was performed in all except three patients. Modified neck dissection was performed in 18 patients (50.0%). In addition, selective neck dissection was performed in 10 patients (27.8%), and extended neck dissection was performed in 3 patients (8.3%). The level of neck dissection was significantly different between the two groups. Concomitant thyroidectomy was performed in 15 patients (41.7%). The mean jejunal flap length was 15.58 cm (range, 10–20 cm). The superior thyroid artery was most frequently used as the recipient artery in 32 patients (88.9%). Additionally, the external jugular vein was used as the recipient vein in 25 patients (69.4%), a branch of the internal jugular vein was used in 8 patients (22.2%), and the vena comitantes of the superior thyroid artery were used in 3 patients (8.3%) (Table 3).

Esophagography was performed in a mean of 18.81 days after surgery (range, 14–27). Oral intake was started a mean of 20.56 days after surgery (range, 14–39 days). Postoperative complications occurred in five patients, and three (7.5%) showed leakage on esophagography, all of whom spontaneously healed with no or limited oral intake and local wound care, frequent suctioning of saliva, and checking of the anastomosis site with laryngoscopy. Hematoma occurred in one case, and another patient experienced flap failure. The failed flap was immediately replaced with a new jejunal flap through take back surgery. Three patients underwent percutaneous endoscopic gastrostomy due to oral intake difficulty in the early postoperative period within two months; one patient showed focal stenosis on endoscopy. Therefore, a balloon dilatation procedure was performed. Donor-site wound dehiscence occurred in two patients, while intussusception was noted in one patient (Table 4).

### 3.2. Questionnaire

Cronbach’s alpha coefficient was 0.881 for the preoperative questionnaire, 0.873 at 3 months (1 month post-PORT) postoperatively, and 0.886 at 12 months postoperatively. Therefore, it showed internal consistency and reliability.

The overall mean preoperative score was 11.19. Improvement in the score was noted at 3 months (1 month post-PORT) and at 12 months postoperatively (mean, 9.58 and 9.44, respectively). The mean preoperative score in the PORT− group was 10.14, which improved to 9.21 and 8.86, respectively, at 3 and 12 months postoperatively. The PORT+ group showed similar results. The mean preoperative score was 11.86, which improved to 9.82 and 9.82, respectively, at 3 and 12 months postoperatively. However, there were no statistically significant differences between the scores at 3 months (or post-radiation 1 month) and 12 months postoperatively in both the PORT+ and PORT− groups, suggesting that postoperative radiation therapy does not have a detrimental effect on the transferred jejunal flap. In addition, the PORT+ and PORT− groups showed no significant difference in scores with time according to the repeated-measures ANOVA test (*p* = 0.310). In contrast, according to the repeated-measures ANOVA test, differences within the PORT+ and PORT− groups with time were significant (*p* < 0.0001). (Table 5, Figure 1)

### 3.3. Risk Factor Analysis

In univariate analysis, no variable was a significant predictor of the score at 12 months postoperatively. Previous radiotherapy and neck dissection had a close statistically significant relationship. The multivariate analysis showed that neoadjuvant chemotherapy, previous radiotherapy, and neck dissection were significant predictors of the score at 12 months postoperatively. The coefficient of determination (R^2^) was 0.413. PORT did not show a significant effect on the 12 months postoperative score (*p* = 0.962) (Table 6).

## 4. Discussion

Complications associated with radiation therapy are difficult to treat and harmful to patients and are thus critical in various fields [7,8,9]. In particular, since complications associated with radiation therapy in head and neck reconstruction can lead to death due to respiratory and dietary system involvement, various surgical methods to prevent and resolve these complications have been studied [10]. Among them, flaps used for pharyngoesophageal reconstruction are not exposed on the outside, and thus, no particular attention has been paid to the changes caused by radiation in these flaps.

Previous studies have mainly focused on surgical techniques aimed at preventing early complications, such as leakage, fistula, and stricture, rather than long-term issues such as side effects of radiation therapy [11,12,13,14,15]. In contrast to these early complications, one of the most important long-term consequences in such patients is the decline in dietary function, which has a profound effect on the quality of life. Many studies have focused on the surgical method in order to minimize the long-term complication rate in terms of fistula, stricture, and leakage [3,12,14]; however, no specific studies focusing on dietary function before and after radiotherapy have been reported.

According to Baglan et al., radiation colitis may occur when the small bowel is exposed to a radiation dose of 45 Gy or more, which causes various symptoms such as proctitis, hemorrhage, fistula, abscess, perforation, and stricture [5]. In particular, chronic colitis cannot be easily resolved. Underlying pathology suggests that fibrin thrombi cause vascular damage and persistent local ischemia due to subintimal thickening of the arteriole, which thereby causes diffuse fibrosis in the lamina propria and submucosa. This diffuse fibrosis additionally triggers vascular damage, creating a vicious cycle that exacerbates local ischemia, resulting in serious complications such as stricture and perforation [16]. The jejunum is a radiosensitive organ; however, the most important factor associated with radiation colitis is the bowel volume being irradiated rather than the radiation dose. Therefore, reconstructed jejunal flaps with relatively low volumes are known to be comparatively safer in terms of preventing radiation colitis. Furthermore, because the reconstructed jejunal flap functions only as a conduit without being involved in the absorption of digested material, it is different from gastrointestinal tract function, which is why jejunal flaps are resistant to radiation [17]. Indeed, late stricture formation after PORT has rarely been reported in previous studies. According to the results of Chan et al., among 82 patients who underwent free jejunal flap transfer, late anastomotic stricture formation occurred in 1 patient in each of the PORT+ and PORT− groups, with no statistically significant difference. Early fistula formation was the only risk factor for late anastomotic stricture formation in this study [1]. In a study of 368 cases by Perez-Smith et al., stricture formation was observed in 10.9% of the patients [18]. Our results are consistent with those of previous studies. Only four patients (11.1%) showed late complications such as stricture formation and swallowing difficulty. We also could not detect any detrimental effect of radiation therapy on the transferred jejunal flap when compared to that in the PORT− group. We speculate that transferred jejunal flaps are not influenced by PORT, and thus, patients undergoing pharyngoesophageal reconstruction with a free jejunal flap may not experience changes in dietary function after PORT.

A free jejunal flap transfer necessitates additional abdominal surgery, such as laparotomy or laparoscopy, for flap harvest. Previous studies have shown that donor site morbidity following abdominal surgery is a disadvantage of the use of jejunal flaps. According to Yu et al., a total of five (15%) cases of adverse events were noted among 31 patients undergoing free jejunal transfer: two cases of bowel obstruction (6%), two of hernia (6%), and one of ileus formation (3%) [3]. However, in our study, donor site complications occurred in only three cases: two cases (5.5%) of wound dehiscence and one case (2.7%) of intussusception. Another study showed that 2 out of 86 patients who received free jejunal flap transfer (2.3%) showed donor site morbidity. This rate was lower than that in the pectoralis major flap group (7.6%) and the free anterolateral thigh flap group (4.2%) [1]. These low rates of donor site morbidity appear to be acceptable once meticulous surgery is performed by an experienced surgeon, and the results of our study are in line with these studies. However, if donor-site morbidity occurs after free jejunal transfer, early detection and appropriate management are needed because it is a potentially life-threatening complication.

Subjective analysis is a widely used modality to analyze swallowing or dietary function [19,20]. For example, the Performance Status Scale for Head and Neck Cancer Patients (PSSHN) is well-known and widely used to assess the functional status of patients with head and neck cancer [21,22]. In our head and neck cancer center, EORTC QLQ-H&N35 was adopted as a generalized assessment to maintain health-related quality of life. For more numerical analysis, examinations such as esophagography or endoscopy could be helpful, but these are performed only in a limited number of patients because they are invasive. In addition, we considered the patient’s self-assessment to be most important in evaluating the patient’s overall dietary function. Therefore, we selected five items that could represent dietary function for the evaluation tool in this study. Of course, this method has a disadvantage because it does not only reflect the radiation effect to the transferred jejunal flap. However, the most important point of view is that the postoperative scores at 3 and 12 months in the PORT+ group were not statistically different, and the pattern of change in the PORT+ group was not significantly different from that in the PORT− group. In addition, in univariate and multivariate analysis, PORT did not show a significant effect on the postoperative score at 12 months. Based on these analyses, we could conclude that the transferred jejunal flap did not show any detrimental effect after PORT.

In the risk factor analysis, preoperative chemotherapy showed a beneficial effect to the dietary function. Preoperative chemotherapy could reduce the extent of tumor and surgery. Therefore, it could induce a beneficial effect to the postoperative function. Radiotherapy is a well-known risk factor in swallowing disorders [23,24,25,26]. Previous radiotherapy showed a detrimental effect on dietary function, which is consistent with the results of previous studies [27,28].

This study had several limitations. This study was based on a retrospective chart review, and thus, the possibility of bias cannot be excluded. Although we attempted to enroll more cases over a long study period, the number of included patients was relatively small because pharyngoesophageal reconstruction itself is rarely performed and long-term follow up is not easy due to the associated mortality rate. This small number of enrolled patients hampered the statistical analysis. The coefficient of determination (R^2^) in the multivariate regression model was 0.413, which indicated a relatively weak effect size. Furthermore, dietary function was subjectively evaluated through a questionnaire, and follow-up esophagography or endoscopic evaluation was only conducted if the patient complained of any difficulty related to dietary function. Therefore, data from an objective examination could not be included. The questionnaires we adopted (EORTC QLQ-H&N35) reflected the patients’ swallowing function, through which we estimated the dietary function. However, it did not reflect the exact type of diet. Therefore, we need a more extensive examination or questionnaire for more exact assessment, which we could not perform due to the retrospective nature of this study.

Nevertheless, we tried to ensure an objective analysis using statistical methods such as Cronbach’s alpha coefficient to evaluate the internal consistency of the questionnaire. As mentioned above, the two major surgical strategies for pharyngoesophageal reconstructions are the transfer of jejunal flaps and fasciocutaneous flaps, such as the anterolateral thigh flap. A comparison of these two methods would be more valuable; however, only a small number of patients undergo fasciocutaneous flap transfer in our institution, thus excluding the possibility of an objective comparison.

## 5. Conclusions

This study shows that PORT does not have a deteriorative effect on free jejunal flap for circumferential pharyngoesophageal defect. Repeated measures ANOVA showed that PORT had no significant impact on dietary function. The multivariate analysis showed that neoadjuvant chemotherapy, previous radiotherapy, and neck dissection were significant predictors of the score at 12 months postoperatively. PORT did not show a significant effect on the 12-month postoperative score. Therefore, free jejunal flap is an effective pharyngoesophageal defect reconstruction method that does not cause any dietary function disruption after PORT.

## Figures and Tables

**Figure 1 jcm-11-02860-f001:**
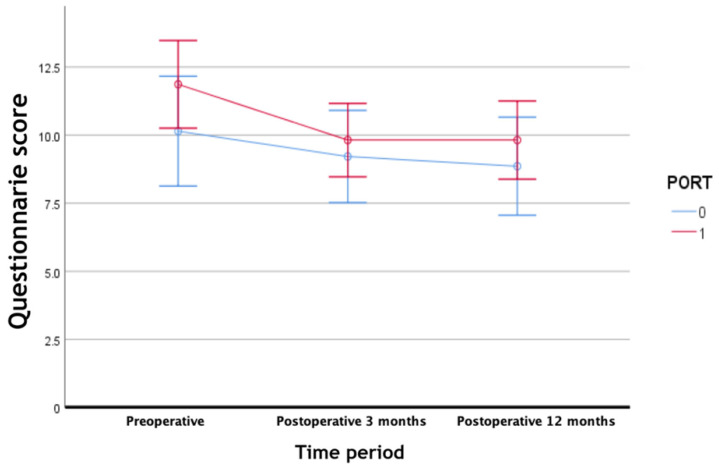
Questionnaire score of the two groups at different time points.

**Table 1 jcm-11-02860-t001:** Five selected items related to dietary function from the European Organization for Research and Treatment of Cancer (EORTC) head and neck cancer module (QLQ-H&N35) [6].

During the Past Week	Not at All	A Little	Quite a Bit	Very Much
Have you had a painful throat?	1	2	3	4
Have you had problems swallowing liquids?	1	2	3	4
Have you had problems swallowing pureed food?	1	2	3	4
Have you had problems swallowing solid food?	1	2	3	4
Have you had trouble eating?	1	2	3	4

**Table 2 jcm-11-02860-t002:** Baseline characteristics of the study population.

Baseline Characteristics	Value	PORT+	PORT−	*p*-Value
Number	36	22 (61.1)	14 (38.9)	
Age at op (yr)	55.22 ± 9.97	54.91 ± 10.94	55.71 ± 8.60	0.817
Sex				0.511
Male	34 (94.4)	20 (90.9)	14 (100)	
Female	2 (5.5)	2 (9.1)	0 (0)	
Follow-up period (mo)	49.47 ± 28.65			
Cancer type				0.754
Hypopharyngeal cancer	28 (77.8)	17 (77.3)	11 (78.6)	
Upper esophageal cancer	4 (11.1)	2 (9.1)	2 (14.3)	
Laryngeal cancer	4 (11.1)	3 (13.6)	1 (7.1)	
Primary or Recurred				0.067
Primary	25 (69.4)	18 (81.8)	7 (50.0)	
Recurred	11 (30.6)	4 (18.2)	7 (50.0)	
Neoadjuvant CTx	11 (30.6)	8 (36.4)	3 (21.4)	0.467
Adjuvant CTx	18 (50.0)	16 (72.7)	2 (14.3)	0.001
Previous RTx	11 (30.6)	5 (22.7)	6 (42.9)	0.273
T stage				0.105
T1	1 (2.8)	0 (0)	1 (7.1)	
T2	5 (13.9)	1 (4.5)	4 (28.6)	
T3	7 (11.1)	6 (27.3)	1 (7.1)	
T4	23 (8.3)	15 (68.2)	8 (57.1)	
N stage				0.216
N0	18 (50.0)	8 (36.4)	10 (71.4)	
N1	11 (30.6)	9 (40.9)	2 (14.3)	
N2	4 (11.1)	3 (13.6)	1 (7.1)	
N3	3 (8.3)	2 (9.1)	1 (7.1)	

CTx, Chemotherapy; RTx, Radiotherapy.

**Table 3 jcm-11-02860-t003:** Baseline characteristics of the study population.

Operation Related	Value	PORT+	PORT−	*p*-Value
Neck dissection				0.04
MRND	18 (50)	14 (63.6)	4 (28.6)	
Selective neck dissection	10 (27.8)	6 (27.3)	4 (28.6)	
Extended neck dissection	3 (8.3)	2 (9.1)	1 (7.1)	
Other	2 (5.5)	0 (0)	2 (14.3)	
None	3 (8.3)	0 (0)	3 (21.4)	
Concomitant Thyroidectomy	15 (41.7)	8 (36.4)	7 (50.0)	0.499
Length of jejunal FF	15.58 ± 3.11	15.45 ± 2.79	15.79 ± 3.66	0.761
Recipient artery				1.000
Sup. Thyroid artery	32 (88.9)	19 (86.4)	13 (92.9)	
Other	4 (11.1)	3 (13.6)	1 (7.1)	
Recipient vein				1.000
Ext. jugular vein	25 (69.4)	15 (68.2)	10 (71.4)	
Branch of internal jugular vein	8 (22.2)	5 (22.7)	3 (21.4)	
Vena comitantes of sup. thyroid artery	3 (8.3)	2 (9.1)	1 (7.1)	

MRND, modified radical neck dissection.

**Table 4 jcm-11-02860-t004:** Outcome variables evaluated.

Variables	Value	PORT+	PORT−	*p*-Value
Esophagography (days)	18.81 ± 2.90	19.05 ± 3.06	18.43 ± 2.68	0.761
Oral intake start (days)	20.56 ± 4.63	21.05 ± 5.47	19.79 ± 2.89	1.000
Early Complication				1.000
Leakage	3 (7.5)	2 (9.1)	1 (7.1)	
Hematoma	1 (2.5)	0 (0)	1 (7.1)	
Flap failure	1 (2.5)	1 (4.5)	0 (0)	
Late complication				1.000
PEG insertion due to swallowing difficulty	3 (7.5)	2 (9.1)	1 (7.1)	
Focal stenosis	1 (2.5)	0 (0)	1 (7.1)	
Donor site complication				1.000
Wound dehiscence	2 (5)	1 (4.5)	1 (7.1)	
Intussusception	1 (2.5)	0 (0)	1 (7.1)	

PEG, percutaneous endoscopic gastrostomy.

**Table 5 jcm-11-02860-t005:** Questionnaire findings and comparison between the groups.

Variables	PORT−	PORT+	Overall
Questionnaire score			
Preop	10.14 ± 3.21	11.86 ± 3.99	11.19 ± 3.76
Postop 3 months	9.21 ± 2.94	9.82 ± 3.22	9.58 ± 3.01
Postop 12 months	8.86 ± 3.35	9.82 ± 3.29	9.44 ± 3.30
*p*-value	0.310 ^†^ <0.0001 ^†^		
Differences within groups with time (*p*-value)			

^†^ Statistically significant values.

**Table 6 jcm-11-02860-t006:** Univariate and multivariate analyses for predictors of dietary function.

Variables	Univariate Analysis		Multivariate Analysis	
	Coefficient	*p*-Value	Coefficient	*p*-Value
Age	0.021	0.715	−0.015	0.728
Sex	1.647	0.170	2.7764	0.170
Cancer type	−0.155	0.959		0.074
Hypopharyngeal cancer	Ref		Ref	
Upper esophageal cancer	0	1	−1.594	0.259
Laryngeal cancer	−0.5	0.773	−2.8434	0.081
Recurred cancer	−1.033	0.395		
Length of jejunal FF	0.223	0.218		
Neoadjuvant CTx	−0.407	0.738	−2.9493	0.006 ^†^
Previous RTx	1.978	0.098	3.4673	0.002 ^†^
Adjuvant CTx	0.0111	0.405		
Postop RTx	0.961	0.212	0.0648	0.962
Early complication	−0.531	0.577		
Neck dissection	0.623	0.069		<0.001
None	Ref		Ref	
MRND	4.778	0.009	7.640	<0.001 ^†^
Selective neck dissection	3.867	0.004	6.448	0.001
Extended neck dissection	3.333	0.162	2.934	0.182
Other	6.667	0.012	8.174	<0.001

FF, free flap; CTx, chemotherapy; RTx, radiotherapy; Postop, postoperative; MRND, modified radical neck dissection. ^†^ Statistically significant values.

## Data Availability

The datasets generated and/or analyzed during the current study are not publicly available since proper ethical permission for open data access has not been obtained but are available from the corresponding author upon reasonable request.

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
