# Peer review of "The Impact of Postoperative Radiotherapy on Dietary Function of Head and Neck Cancer Patients after Pharyngoesophageal Reconstruction with Free Jejunal Flap"

_jcm, 2022, doi:10.3390/jcm11102860_

Round 1
Reviewer 1 Report
This is an interesting study about the impact of postoperative radiotherapy on dietary function of head and neck cancer patients after pharyngoesophageal reconstruction with free jejunal flap. The authors evaluated 36 patients.
The paper is well written. However, some issues remain.
The authors stated that the superior thyroid artery was most frequently 107 used as the recipient artery in 36 patients (88.9%). Since the sample consisted of 36 patients, the absolute number was not correct.
Please better describe local wound care for leakages.
I think that the patient with flap failure should be excluded from the study.
Author Response
Response to reviewers
We thank the editor, and all reviewers for reviewing our paper in detail. We have revised our manuscript according to your advice and comments. The revised parts are indicated in red font. We have provided point-by-point responses to each reviewer’s comments.
<Reviewer 1>
This is an interesting study about the impact of postoperative radiotherapy on dietary function of head and neck cancer patients after pharyngoesophageal reconstruction with free jejunal flap. The authors evaluated 36 patients.
The paper is well written. However, some issues remain.
The authors stated that the superior thyroid artery was most frequently 107 used as the recipient artery in 36 patients (88.9%). Since the sample consisted of 36 patients, the absolute number was not correct.
: Thank you for your helpful comments. This mistake was corrected and we additionally checked the all statistical values, in which several errors were found and corrected.
Please better describe local wound care for leakages.
: Thank you for your helpful comments. In fact, no or limited oral intake was the most important management method and supportive for local wound care. We have described the details of the local wound care in line 118.
I think that the patient with flap failure should be excluded from the study.
Thank you for your helpful comments. We also had considered excluding flap failure cases. As you know, the ischemic time of the jejunal flap is very short, and salvage is almost impossible. Therefore, we abandoned the salvage procedure and replaced the flap immediately through take-back surgery. This means that the patients underwent surgery twice within a short period, which could have affected the surgical results. Therefore, we did not remove flap failures from our analysis. The details of flap failure have been added to line 119-120.

Reviewer 2 Report
These authors examined the impact of postoperative radiotherapy (PORT) on dietary function in head and neck cancer patients after pharyngoesophageal (PE) reconstruction with free jejunal flap. This study is novel and adds to the literature on outcomes after head and neck cancer treatment. The writing is clear, organized, and concise. The lit review is appropriate. The lit review should include references concerning swallowing and dietary outcomes post-PE reconstruction with free jejunal flap, particularly since this study is examining diet outcomes after surgery. There are data showing the impact of this type of surgery on pharyngeal phase swallow functioning, per my comment below. Methodology is clear, as are stats. Results, including tabs and figures, are clear. Discussion is thoughtful, including limitations. A few comments:
- It appears that none of the patients underwent a total or partial laryngectomy, despite 8 patients having a diagnosis of laryngeal cancer. Please clarify.
- The EORTC QLQ-H&N35 is a PRO measure that assesses a patient’s perception of their difficulty swallowing, not really “dietary function”. The 5 questions they used do address different liquid/food consistencies. However, responses do not mean that the patients are eliminating these liquid/food consistencies from their diet. There is no mention as to type of diet that these patients were able to take post-surgery. The authors mention the PSSHN in their discussion, which is a tool that specifies the exact type of diet the patients are eating. This second tool could easily have been administered to determine what diet the patients were on. The authors do note that lack of additional questionnaires was a limitation to their study.
- The authors note that 3 patients underwent PEG placement due to “oral intake difficulty”. How was oral intake/oral function assessed? Clinical/bedside evaluation? Instrumental assessment? The authors note that esophagography and endoscopic evaluation were used when difficulty swallowing was identified. However, were any tools to examine the pharyngeal phase function of swallowing used, as free jejunal reconstruction has been found to tether the larynx during swallowing and result in slowed airway entrance closure, reduced laryngeal motion, reduced bolus clearance, reduced PE segment opening and post-surgery. Please elaborate.
- Did any of these patients undergo swallow therapy? please comment.
Reviewer 3 Report
You have investigated the effect of radiotherapy on food intake in patients with pharyngoesophageal defect reconstruction using a free jejunal flap. The research perspective is of interest to the reader. However, there is a problem with the methodology of the study. The first is the misuse of statistical analysis, and the second is the interpretation of the results.
1. Misuse of statistical analysis
Even though this study uses U-tests for between-group comparisons, ANOVA was selected.
Did you check for normality? Using ANOVA for nonparametric data is inappropriate.
Also, you are conducting multiple regression analysis in a small sample size study, but you are inserting more variables than the sample size limit, right?
Can you provide the coefficient of determination R2? It is suggested that it is probably very low.
2. interpretation of the results
You conclude that PORT has no effect on dietary intake, which is inappropriate.
Even though this study is not a non-inferiority study design, it is misleading of the results to conclude that the conclusion is no different. The conclusion is "we don't know if there is a difference".
Considering the methodological issues and the misleading conclusion, the quality of this study would make the paper unacceptable and would require a significant amount of time to improve.
Thank you very much for your time.
Author Response
Response to reviewers
We thank the editor, and all reviewers for reviewing our paper in detail. We have revised our manuscript according to your advice and comments. The revised parts are indicated in red font. We have provided point-by-point responses to each reviewer’s comments.
<Reviewer 3>
You have investigated the effect of radiotherapy on food intake in patients with pharyngoesophageal defect reconstruction using a free jejunal flap. The research perspective is of interest to the reader. However, there is a problem with the methodology of the study. The first is the misuse of statistical analysis, and the second is the interpretation of the results.
- Misuse of statistical analysis
Even though this study uses U-tests for between-group comparisons, ANOVA was selected.
Did you check for normality? Using ANOVA for nonparametric data is inappropriate.
: Thank you for your helpful comments. We used repeated-measures ANOVA method. As you commented, ANOVA follows the assumptions of normality and equal variance. However, repeated-measures ANOVA tends to be relatively flexible under the assumption of normality. Even in textbooks, a comparison between two groups of seven subjects is presented as an example. In repeated-measures ANOVA, each measurement is not independent, but has an intra-individual correlation. Therefore, it follows the assumption of sphericity instead of equal variance. The assumption of equality, which means that the structure of the variance is the same between groups, is added.
In Mauchly's sphericity test, which indicates the assumption of sphericity in our analysis, the p-value was < 0.0001. Therefore, we adopted the Greenhous-Geisser and Huynh-Feldt correction values.
In Box's identity test, which indicates the assumption of identity, the p-value was 0.159. Therefore, we confirmed that the distribution structures of the two groups were identical.
Also, you are conducting multiple regression analysis in a small sample size study, but you are inserting more variables than the sample size limit, right?
: Thank you for your helpful comments. As you mentioned, our study population was too small to include many variables in the multiple regression analysis. There was a description error in Table 6. In multiple regression analysis, only six variables were included: age, sex, neoadjuvant CTx, previous RTx, posted RTx, cancer type, and neck dissection. We have corrected Table 6 accordingly.
Can you provide the coefficient of determination R2? It is suggested that it is probably very low.
: Thank you for your helpful comments. The coefficient of determination R2 was 0.4132. As expected, this value was relatively low. We have added more description of the R2 value and its meaning in lines 151 and 239-240.
- interpretation of the results
You conclude that PORT has no effect on dietary intake, which is inappropriate.
Even though this study is not a non-inferiority study design, it is misleading of the results to conclude that the conclusion is no different. The conclusion is "we don't know if there is a difference".
Considering the methodological issues and the misleading conclusion, the quality of this study would make the paper unacceptable and would require a significant amount of time to improve.
Thank you very much for your time.
: Thank you for your helpful comments. We analyzed the effect of PORT after pharyngoesophageal reconstruction using a free jejunal flap. This was undertaken because there is an opinion that a jejunal free flap may be more vulnerable to PORT than a skin flap such as ALT or radial forearm. Since there was no control group with a skin flap, we tried our best to proceed with the study under the given conditions. Therefore, we compared patients who did not receive PORT after the jejunal flap in various ways and confirmed that PORT did not have particularly deleterious effect on patients with the jejunal flap. We did not conclude that PORT had no effect on dietary intake, but it did not have a deleterious effect on the transferred jejunal flap.
Nevertheless, we fully agree with your opinion that our argument may have been distorted. Especially, the last sentence in conclusion “Therefore, free jejunal flap is an effective pharyngoesophageal defect reconstruction method that does not cause any dietary function disruption after PORT.” could be regarded as an overstatement. Therefore, we deleted this sentence and hope that you understand the interpretation of our analysis.

Round 2
Reviewer 3 Report
Thank you for your thoughtful and logical response.
During the initial peer review, I chose to reject the paper because I thought it would take a significant amount of time to revise the paper. However, you have written a logical response letter, which has resolved my doubts.
I believe this paper adequately meets the criteria to be accepted.